# Factors Influencing Young Korean Men’s Knowledge and Stigmatizing Attitudes about HIV Infection

**DOI:** 10.3390/ijerph17218076

**Published:** 2020-11-02

**Authors:** Mi-So Shim, Gwang Suk Kim

**Affiliations:** Mo-Im Kim Nursing Research Institute, College of Nursing, Yonsei University, Seoul 03722, Korea; misoshim1111@gmail.com

**Keywords:** HIV, knowledge, attitude, stigma, influencing factor

## Abstract

Stigma against people living with HIV (PLHIV) fosters depression and negatively impacts the quality of life in PLHIV and is a barrier to the whole process of treatment. This study aimed to identify the levels of knowledge and stigmatizing attitudes toward HIV infection among Korean men in their 20s and the factors influencing them. A cross-sectional design was used. Two hundred and eight Korean men in their 20s responded to self-report questionnaires that included items on knowledge and stigmatizing attitudes about HIV infection, subjective norms for safer sexual behaviors (SSBs), participants’ HIV-related characteristics, sex-related characteristics, and general characteristics. The mean score (±SD) for knowledge was 13.9 (±5.28) and for stigmatizing attitudes was 64.1 (±11.42). In quantile regression analysis, exposure to HIV-related information was a significant factor for knowledge in every quantile, and experience of HIV education was a significant factor in the 50th quantile. Experience of meeting PLHIV was a significant factor for stigmatizing attitudes in the 25th quantile, and subjective norms for SSB were a significant factor for stigmatizing attitudes in the 25th and 50th quantiles. Findings suggest the need for intervention to improve young Korean men’s knowledge, as well as intervention focusing on norms for SSB, to prevent stigmatizing attitudes.

## 1. Introduction

HIV-related stigma refers to negative beliefs, feelings, and attitudes toward people living with HIV (PLHIV) and acts as a major barrier to HIV prevention and treatment [1]. HIV stigma and discrimination can drive PLHIV to recurring depression and even suicide [2] and have adverse effects on their quality of life [3]. In Korea, people tend to hold more explicitly stigmatizing attitudes toward PLHIV than in many other countries, underlying which a variety of factors come into play: fewer opportunities for personal contact with PLHIV due to low prevalence [4], attribution of responsibility for HIV infection to PLHIV themselves, for example, due to their presumed careless sexual relationships [5], and lack of knowledge about transmission routes, contagiousness, and positive prognosis of HIV [6]. HIV stigma and discrimination trigger fears of HIV infection and undermine the efforts of PLHIV to seek access to services for HIV prevention, treatment, and care [1]. It is therefore essential to put effort into reducing HIV-related stigma and discrimination in order to strengthen HIV prevention and create a more therapeutic environment for PLHIV.

As of 2019, the number of PLHIV in Korea is 13,857, with the number of persons with newly diagnosed HIV infection remaining steady: 1008 in 2017, 989 in 2018, and 1005 in 2019 [7]. In 2019, PLHIV in their 20s to 40s accounted for 62.8% of Korean PLHIV; of new diagnoses, a plurality was in their 20s (35.8%, *n* = 438), followed by 30s (27.9%, *n* = 341) and 40s (16.5%, *n* = 202), together making up 80.3% of the total [7]. As for gender, men account for 93.3% of the total number of PLHIV (12,926 out of 13,857); as for routes of HIV transmission in 2019, sexual contact is the main exposure route, at 81.7% (heterosexual contact: 37.7%, homosexual contact: 44.0%)—potentially much higher, considering the high percentage of no responses (18.1%) [7]. Given the steady occurrence of newly diagnosed HIV infection in Korea and the high rate of newly diagnosed young males, there is a compelling need to assess knowledge about and stigmatizing attitudes toward HIV infection among these men and provide them with basic information and intervention to improve their knowledge and attitudes.

According to the 2015 national survey conducted by the Korea Centers for Disease Control and Prevention (KCDC) regarding HIV/AIDS-related knowledge, attitudes, and beliefs, the words most frequently associated with AIDS include “incurable disease” and “death” (25.3%), followed by “fear,” “horror,” and “danger” (11.5%). Alongside these negative views, the general population in Korea has low knowledge, such as transmission routes and the fact that treatment allows people with HIV to live healthy lives [6]. For the assessment of knowledge about HIV/AIDS, UNAIDS (2009) recommended the following five items: “Is it possible for a healthy-looking person to have the AIDS virus?” “Can people reduce their chance of getting the AIDS virus by using a condom every time they have sex?” “Can people reduce their chance of getting the AIDS virus by having just one uninfected sex partner who has no other sex partner?” “Can people get the AIDS virus by sharing food with a person who has AIDS?” and “Can people get the AIDS virus from mosquito bites?” [8]. Korean studies have indeed used these five items. With these questions alone, however, one cannot assess knowledge conducive to positive perception of and attitudes toward HIV infection, such as that not all PLHIV have AIDS, the low likelihood that PLHIV under medication will transmit HIV through sexual contact, and that life expectancy is not shorter than that of the general population. Korean studies assessing knowledge of these recent trends are lacking.

In Korea, public campaigns and education activities are being promoted to the public as a major effort to improve knowledge about and attitudes toward HIV infection [6], but there is a lack of systematic education and experts to carry out these efforts [9]. The standard for sex education in Korean schools mentions HIV infection as a sexually transmitted disease, but the content is limited to HIV prevention [10]. Stigmatizing attitudes toward HIV infection in Korea also affect the public perception of homosexuality. In 2004, a survey of the gender identity of 1613 men living in Seoul, Republic of Korea revealed that 0.2% of men identified themselves as homosexual and 0.3% as bisexual, while only 1.1% had homosexual experiences [11]. According to the results of one study, in Korea, 70% of respondents answered that same-sex relationships are “always wrong,” while only 7% answered “not wrong at all” [12]. With a considerable proportion of routes of HIV infection transmission reported to be homosexual contacts in Korea [7], HIV infection has often been criticized as a “homosexual disease.” In this context, to evaluate the factors influencing knowledge and attitudes about HIV infection, it is necessary to consider people’s educational experience around HIV infection and their norms for sexual behavior.

Whereas specific sociodemographic characteristics have been presented as factors influencing HIV-related knowledge and attitudes [5,13], little research has investigated socio-psychological factors that may serve as basic data necessary for developing intervention programs designed to improve HIV-related knowledge, attitudes, and stigma and thus foster testing, early therapy onset, and prevention. Given the high proportion of men in their twenties among persons with newly diagnosed HIV, it is especially necessary to assess HIV-related knowledge, stigmatizing attitudes, HIV-related experiences, and sexual behavior among men in their 20s. This study aimed to assess HIV-related knowledge and stigmatizing attitudes among Korean men in their 20s and to determine the influencing factors.

## 2. Materials and Methods

### 2.1. Study Design and Participants

This was a cross-sectional study. The participants were recruited by convenience sampling from registrants with the online survey firm “dataSpring.” DataSpring is a survey platform that provides Asian panels, including Korean panels [14]. In 2018, in the Korean dataSpring research panel, there were 125,745 men, of which 32.7% were in their 20s. Inclusion criteria were (1) men in the age group 20–29 living in Korea, (2) capable of reading and answering the questionnaire, and (3) agreeing to participate in the study. Exclusion criteria were (1) people diagnosed with HIV infection, (2) health care employees or professionals, and (3) people involved in HIV-related organizations. Among the panels provided by dataSpring, only males living in Korea aged 20–29 who met the inclusion criteria for this study were extracted. Prior to the online survey, a questionnaire for screening was presented, and the respondents corresponding to the exclusion criteria were excluded from participation in the study.

The sample size was calculated using G*Power 3.1.9.2. (Heinrich-Heine-Universität, Düsseldorf, Germany) Adequate sample size for linear multiple regression was computed to yield 184, according to the following criteria: effect size = 0.15, significance level = 0.05, statistical power = 0.95, and predictive factors = 12 (*n*). Since the needed sample size for quantile regression is 100 or more [15], the sample in this study (*n* = 208) was sufficient for analysis.

### 2.2. Measures

#### 2.2.1. HIV-Related Knowledge

We used the HIV/AIDS knowledge scale developed by Prati et al. (2016), translated into Korean with permission [16]. The original tool consists of 29 items designed to check the knowledge about routes of HIV transmission, prevention, treatment, and prognosis. A score of 1 (correct answer) or 0 (wrong or “don’t know”) is allocated to each item. After deleting two items that had similar content, 27 items were used in this study. The original tool and the modified tool had McDonald’s omega = 0.94 and Cronbach’s alpha = 0.81, respectively.

#### 2.2.2. HIV-Related Stigmatizing Attitudes

We used the tool developed by Beaulieu, Adrien, Potvin, and Dassa (2014), translated into Korean with permission [17]. This tool consists of 27 items: (1) concerns about occasional encounters (3 items), (2) avoidance of personal contact (3 items), (3) responsibility and blame (6 items), (4) liberalism (4 items), (5) non-discrimination (5 items), (6) confidentiality of serological status (3 items), (7) criminalization of HIV transmission (3 items). Each item is rated on a 4-point Likert scale from 1 (strongly disagree) to 4 (strongly agree), with reverse scoring on some items. The higher the total score, the less stigmatizing the attitudes toward HIV infection. The original tool and the one used in this study had Cronbach’s alpha = 0.88 and 0.86, respectively.

#### 2.2.3. HIV-Related Characteristics

Eight items were included assessing HIV-related characteristics: presence/absence of PLHIV among family members, friends, colleagues, and acquaintances; experience meeting and talking with PLHIV; experiences of receiving HIV-related education (in school, work, etc.), HIV testing, counseling, or exposure to HIV-related information (through TV programs, internet, social networking sites, etc.); sources of HIV/AIDS-related information; etc.

#### 2.2.4. Subjective Norms for Safer Sexual Behavior

For assessment of these norms, we used a tool developed and modified by Ji (1993) [18] and Lee and Chon (2006) [19], respectively, and tailored to safer sexual behavior (SSB) by Kim (2014) [20], with permission. This tool consists of two items: one assessing intention to comply with the opinions of influential persons and the other assessing normative beliefs of people around participants regarding safer sexual practices. Compliance intention was measured on a 4-point Likert scale (0 = don’t know/a little; 1 = somewhat; 2 = generally; 3 = a great deal) and normative beliefs were measured on a 7-point Likert scale (3 = very much so; 0 = don’t know; −3 = not at all). The total score for subjective norms for SSB was calculated by multiplying the scores of these two items (range: −9 to +9); the higher the score, the higher the pressure from friends and acquaintances to adhere to safer sexual practices.

#### 2.2.5. Sex-Related Characteristics

Two items were included to assess sex-related characteristics: sexual orientation and sexual experience.

#### 2.2.6. General Characteristics

General characteristics consisted of five items: age, marital status, education level, economic status, and employment status.

### 2.3. Data Collection

This study was approved by the Institutional Review Board of the authors’ institution (project number: Y-2018-0037). Data were collected from 17 May to 8 June 2018, via an online self-report questionnaire survey. A total of 208 questionnaires were collected, and all of them were included in the analysis. It took about 15 min to complete the questionnaire, and the respondents were given a token (a monetary point) of appreciation via the online survey company.

### 2.4. Data Analysis

The data were analyzed using SPSS WIN 23.0 (IBM Corp., Armonk, NY, USA) and STATA/SE 15.0. (Stata Corp., College Station, TX, USA) We performed a t-test and ANOVA for analysis of differences in HIV-related knowledge and stigmatizing attitudes by participant characteristics, and Pearson’s correlation to determine correlations among the variables. Last, quantile regression was performed to identify factors influencing participants’ knowledge and stigmatizing attitudes by quantile. Using quantile regression, introduced by Koenker and Bassett (1978) [21], the regression coefficients of the independent variables were estimated quantile by quantile using the conditional quantile function, unlike the classical regression model based on the conditional mean function [22].

## 3. Results

The participants’ scores for knowledge ranged between 0 and 25 (mean: 13.9 ± 5.28), stigmatizing attitudes between 35 and 108 (mean: 64.1 ± 11.42), and subjective norms for SSB between −6 and 9 (mean: 1.8 ± 2.63). Table 1 presents the characteristics of the participants and the characteristic-dependent differences in their HIV-related knowledge and stigmatizing attitudes. In the case of sexual orientation, heterosexual was the most common, at 183 (88.0%), followed by bisexual 12 (5.8%), homosexual 5 (2.4%), asexual 3 (1.4%), and others 5 (2.4%). The results of the t-test and ANOVA of differences in HIV-related knowledge and stigmatizing attitudes by participants’ characteristics yielded the following findings: knowledge about HIV infection showed significant differences by experience of receiving HIV-related education (t = 3.156, *p* = 0.002), non-voluntary HIV testing (t = 3.994, *p* < 0.001), and HIV-related information (t = 3.756, *p* < 0.001); none of the characteristics showed statistically significant differences in stigmatizing attitudes toward HIV infection, although non-significant differences were observed depending on employment status (t = −1.910, *p* = 0.057), experience of meeting PLHIV (t = 1.848, *p* = 0.066), and sexual orientation (t = −1.788, *p* = 0.075).

Table 2 outlines the percentages of correct answers for the 27 items of the knowledge assessment tool. The participants scored low on the items “HIV-positive persons who are on regular antiretroviral treatment (ART) and responding to treatment are less likely to transmit HIV” (31.7%), “HIV is transmitted from mother to child during pregnancy” (32.2%), “HIV can be diagnosed three months after exposure at the earliest” (32.7%), and “There is no significant gap in life expectancy between PLHIV on regular ART [antiretroviral therapy] and the general population” (35.6%). The two items regarding the routes of HIV transmission in Korea resulted in correct scores as low as 26.4% and 24.5%, respectively.

Table 3 outlines the results of correlation analysis among participants’ HIV-related knowledge, stigmatizing attitudes, and subjective norms for SSB. Whereas no statistically significant correlations were observed between HIV-related knowledge and overall stigmatizing attitudes, significant correlations were found between HIV-related knowledge and five of seven subcategories of stigmatizing attitudes, namely concerns about occasional encounters (r = −0.219), liberalism (r = 0.222), non-discrimination (r = −0.241), confidentiality of serological status (r = −0.187), and criminalization of transmission (r = −0.239). Subjective norms for SSB showed statistically significant correlations with HIV-related knowledge (r = 0.188) and stigmatizing attitudes (r = −0.183).

Table 4 presents the results of the linear regression and quantile regression of variables, showing significant differences (*p* < 0.10) in HIV-related knowledge and stigmatizing attitudes by participants’ characteristics based on the t-test and ANOVA. From the linear regression, experience of non-voluntary HIV testing (coef. = 1.87, *p* = 0.033), exposure to HIV-related information (coef. = 2.37, *p* = 0.006), and subjective norms for SSB (coef. = 0.30, *p* = 0.026) were derived as factors influencing HIV-related knowledge. Quantile regression yielded the following findings: in the quantile with low HIV-related knowledge (25th), knowledge score was significantly higher in the group that had been exposed to HIV-related information (coef. = 2.66, *p* = 0.016); in the medium-knowledge quantile (50th), exposure to HIV-related information (coef. = 1.51, *p* = 0.039) and experience of HIV-related education (coef. = 2.56, *p* = 0.007) were significantly influential; in the high-knowledge quantile (75th), exposure to HIV-related information (coef. = 2.29, *p* = 0.037) was significantly influential.

No statistically significant influential factors were derived from the linear regression of participants’ stigmatizing attitudes toward HIV. Quantile regression, however, yielded the following findings: in the quantile with low scores on stigmatizing attitudes (that is, more stigma) (25th), experience meeting PLHIV (coef. = 18.15, *p* = 0.048) and subjective norms for SSB (coef. = −1.17, *p* = 0.006) were derived as factors influencing stigmatizing attitudes; in the medium-stigma quantile (50th), subjective norms for SSB were again significantly influential (coef. = −1.00, *p* = 0.003); however, in the high-stigma quantile (75th), no significant influential factors were derived.

## 4. Discussion

In this study, a cross-sectional survey was conducted on 208 Korean men in their 20s, and their level of knowledge and stigmatizing attitudes toward PLHIV and the factors affecting them were identified. The results show that young Korean men had low knowledge of the prognosis and transmission routes of HIV infection and high stigmatizing attitudes toward HIV infection. Non-voluntary HIV testing, exposure to HIV-related information, experience of HIV education, and SSB were identified as factors affecting knowledge about HIV infection, and experience meeting PLHIV and SSB were identified as factors affecting stigmatizing attitudes.

Respondents’ mean knowledge score was 13.9 (±5.28) out of 25, with conspicuously low scores for the items on the positive prognosis of HIV infection and transmission routes. Of five items recommended by UNAIDS (2009) for HIV-related knowledge assessment, on the item for HIV transmission by mosquito bite, only 36.1% of answers were correct, lower than in the prior survey of the general population (47.7%) [6]. The lack of HIV-related knowledge is the major factor for negative attitudes toward PLHIV [23,24,25], and knowledge about the positive prognosis of HIV infection and treatment for prevention is a crucial element in strategies for destigmatizing HIV [16]. Moreover, the lack of knowledge about routes of HIV transmission can cause prejudice against PLHIV by triggering groundless fears of infection. Therefore, treatment-based positive prognosis and information about routes of HIV transmission need to be included in intervention programs for improving HIV-related knowledge and reducing stigmatizing attitudes among men in their 20s.

Analysis of factors influencing participants’ HIV-related knowledge revealed that the group that had been exposed to HIV-related information scored higher than the group that had not. According to the informatics model, knowledge, information, and data are located at the top, middle, and bottom in a pyramid-shaped hierarchy [26]. In this model, simply collecting and presenting data does not provide useful information and knowledge; it becomes knowledge through complex processes such as induction and deduction [26]. Therefore, in order to improve knowledge of HIV infection, efforts to help form knowledge by generating useful information through appropriate processing of data are needed. Meanwhile, respondents with exposure to HIV-related information were asked to indicate its sources (multiple selections allowed); the most frequent source was TV news/current affairs programs (56.5%), followed by TV culture/documentary programs (45.7%) and Internet news sites (34.8%). A Chinese study similarly reported that mass media are the major source of HIV-related information and noted that exposure to mass media is positively associated with knowledge about routes of HIV transmission [27]. Thus, mass media should be used strategically to deliver accurate information and eliminate prejudices and discrimination against PLHIV [28]. Kerr et al. (2015) reported positive effects of media on improving HIV-related knowledge and destigmatizing HIV in a study with African-American adolescents in which a school program for improving knowledge about HIV transmission and prevention was accompanied by media intervention (TV and radio) [29]. In Thailand, a community-based intervention program was implemented to improve HIV-related knowledge and destigmatize HIV by delivering HIV-related education and information services as edutainment, via radio dramas and the like [30].

In the medium-level HIV-related knowledge quantile (50th), the knowledge score was higher in the group that had been exposed to HIV-related education than in the group that had not, therefore, providing school education can be a good way to increase knowledge about HIV infection. In Korea, nationwide sex education standards adopted in 2015 provide a framework for systematic, continuous sex education in schools, including education contents on HIV infection and general transmission and prevention of sexually transmitted diseases beginning in higher elementary school grades [10]. However, HIV and AIDS material set out in sex education standards is limited to transmission routes and prevention [10]. To help destigmatize HIV, content is also needed on the positive prognosis of PLHIV on regular ART. Furthermore, Lee (2016) noted problems interfering with school-based sex education, such as allocation of class hours, resource development and distribution, and insufficient teacher training and skills communicating information on sexual issues, and highlighted the need for institutional policy support to address such lacks [9].

Linear regression results showed that participants with experience of HIV testing (albeit non-voluntarily) scored higher in HIV-related knowledge. In another study, as well, HIV testing was a significant positive influential factor for knowledge and stigma [31]. That is, experience of even non-voluntary HIV testing enhances HIV-related interest and knowledge. In 2006, the U.S. Centers for Disease Control and Prevention (CDC) recommended integrating HIV testing into routine medical examination to strengthen early diagnosis and prevention of transmission [32], as well as to facilitate consultation with health care providers and create a general atmosphere of acceptance of HIV testing in not only high-risk populations but also the general population [33]. In addition, the U.S. Preventive Services Task Force recommends HIV screening for adolescents and adults aged 15 to 65 years along with pregnant women [34]. In Korea, HIV screening is mandatory for “persons engaged in businesses requiring frequent contacts with the public” under Article 8 of the AIDS Prevention Law, and HIV testing is part of routine prenatal testing. With reference to the U.S. example in routine testing strategy, discussion of guidelines and institutional policies to popularize HIV testing and overcome public resistance to HIV screening, taking into account the Korean context, is needed in Korea. However, if HIV testing becomes part of routine medical examination, ethical issues will arise, such as disclosure of HIV test results, confidentiality, and psychological and emotional stress in HIV-positive cases [33]. To address such problems, strategies may include providing sufficient pre-test explanation and setting up counseling and treatment networks for persons newly diagnosed with HIV.

The mean score for stigmatizing attitudes among Korean men in their 20s was 64.1 (±11.42), much lower (more stigmatizing) than those of the Canadian general population (80.9, *n* = 1500) and of U.S. college students (74.6, *n* = 2343) obtained using the same tool [28,31]. HIV prevention campaigns in Korea in the 1980s and 1990s focused on transmission prevention and resorted to a fear appeal strategy [35]; this can result in unintended stigmatization of HIV [36]. Indeed, people exposed to HIV-related information through such campaigns had more negative attitudes toward HIV than people exposed to HIV-related information through TV culture or documentary programs [35]. This indicates that such TV programming can serve as an indirect channel to alleviate prejudices against PLHIV by presenting their normal daily lives, contradicting stigmatizing views of them [35]. In particular, “undetectable = untransmittable (U = U)” or “treatment as prevention” views, which entail that PLHIV who have a suppressed viral load do not spread the virus to others, are being emphasized [37]. In Korea, efforts are also being made to spread such campaigns around PLHIV communities [38]. Thus, programs for destigmatizing HIV should use TV programming to try to counter negative depictions and represent HIV as a manageable, chronic disease.

In the present study, in the quantile (25th) where attitudes toward HIV infection among men in their 20s were most negative, the influential factors were experience of meeting PLHIV and the subjective norms for SSB. If they had no experience of meeting PLHIV or their subjective norms for SSB were higher, the more stigmatizing the attitudes toward HIV infection were. The nature of HIV infection makes it vulnerable to stigmatization, serving as a rationale for the accusation that HIV is transmitted by abnormal sexual practices such as homosexuality or promiscuity; it is assumed that the higher one’s subjective norms for SSB, the more stigmatizing one’s attitude toward HIV infection [39]. In countries with low HIV infection prevalence, including Korea, where most people encounter PLHIV only rarely, they are likely to learn to hold stigmatizing attitudes toward PLHIV in the course of indirectly learning about HIV through mainstream or social media or hearsay [4]. For example, after the first reports of HIV infection among gay men in the U.S. in the early 1980s, mainstream media began to convey an intermingled picture of homosexuality, HIV infection, and AIDS, giving rise to HIV stigma based on the general perception that HIV infection is a “gay disease” [40]. To reduce HIV stigma, Bos, Schaalma, and Pryor (2008) proposed providing proper information about HIV infection and, concurrently, giving people opportunities to meet PLHIV or to become aware that there are PLHIV among “regular” people living normal lives [41]. In the U.S., the 1991 case of basketball star Earvin “Magic” Johnson, who publicly declared his HIV infection due to heterosexual transmission, contributed greatly to correcting the myth that HIV transmission is limited to the gay population and fostering awareness that anybody can contract HIV, leading to an increase in the number of people undertaking HIV screening [42]. This is a good example of the destigmatizing potential of offering positive personal contacts with PLHIV, whether with beloved public figures or face-to-face with “regular people.”

On the other hand, our correlation analysis revealed that the higher the knowledge score, the more negative the results on subcategories of stigmatizing attitudes: concerns about occasional encounters, non-discrimination, confidentiality of serological status, and criminalization of transmission. This may be explained by the low rate of correct answers of knowledge items regarding positive prognosis and transmission routes, which are important information for the formation of less negative attitudes toward HIV infection. This highlights the need to destigmatize HIV through education emphasizing the dramatically improved prognosis of PLHIV and proper knowledge about transmission routes, making it clear that PLHIV on a regular treatment regimen are at low risk of transmitting HIV through sexual contact and that HIV is not transmitted by, for instance, food sharing or mosquito bites.

As for limitations, first, as a cross-sectional study, this research cannot identify cause–effect relationships. Second, the assessment tools used in this study to quantify the main dependent variables—HIV-related knowledge and stigmatizing attitudes—were not developed in Korea but translated by the author, and their validity and reliability are unclear. Therefore, further research is needed to validate these assessment tools in the future. Third, since the participants were recruited by convenience sampling, appropriate care should be taken in interpreting and generalizing the study results. In the future, it will be necessary to conduct large-scale research using random sampling methods to garner basic data on policies that can improve knowledge and attitudes about HIV infection.

## 5. Conclusions

This study was conducted to determine factors influencing HIV-related knowledge and stigmatizing attitudes among Korean men in their 20s by level and quantile. Respondents had low knowledge on the improving prognosis of HIV infection and on transmission routes and negative, stigmatizing attitudes toward HIV infection. From the linear regression of factors influencing HIV-related knowledge, it was found that experience of non-voluntary HIV testing, exposure to HIV-related information, and subjective norms for SSB were significant. From quantile regression, exposure to HIV-related information was derived as a common factor in all quantiles. In the quantile of 50%, experience of HIV education was a significant factor in HIV-related knowledge. In the quantile where stigmatizing attitudes toward HIV infection were strongest (25th), experience of meeting PLHIV was derived as an influential factor. Furthermore, lower subjective norms for SSB were an influential factor in the quantiles where stigmatizing attitudes were 25% and 50%.

Therefore, it is necessary to develop intervention programs providing more access to HIV-related information and education to improve the level of HIV-related knowledge among Korean men in their 20s and to provide policy support at the institutional level to encourage the general population to undertake HIV screening. When offering interventions to reduce stigmatizing attitudes, particular attention should be given to the choice of contents and methods to prevent indirect learning of stigmatizing attitudes; this should include providing opportunities for positive personal meetings with PLHIV and ensuring accurate perception of the HIV-positive status. Thus, the results of this study can provide basic data for future work aiming at interventions to improve HIV-related knowledge and reduce the stigmatizing attitudes among Korean men in their 20s, who show widely varying levels of HIV-related knowledge and HIV stigma.

## Figures and Tables

**Table 1 ijerph-17-08076-t001:** Differences in knowledge and stigmatizing attitudes about HIV infection by characteristics of participants (*n* = 208).

Variables	Categories	*n* (%)	Knowledge		Stigmatizing Attitude	
			Mean (SD)	t/F	Mean (SD)	t/F
Age	20–24	87 (41.8)	13.6 (5.30)	−0.813	64.5 (11.00)	0.398
	25–29	121 (58.2)	14.2 (5.27)		63.8 (11.75)	
Marital status	Unmarried	196 (94.2)	13.8 (5.30)	−1.283	64.2 (11.57)	0.528
	Married	12 (5.8)	15.8 (4.80)		62.4 (8.87)	
Education level	≤High school	35 (16.8)	12.0 (6.60)	3.019 *	64.0 (10.85)	0.027
	In college	81 (38.9)	14.1 (5.09)		63.9 (11.31)	
	≥College	92 (44.2)	14.5 (4.75)		64.3 (11.84)	
Economic status	≤Mid-to-low	84 (40.4)	14.8 (4.71)	1.919 *	62.6 (11.20)	−1.606
	≥Mid-to-high	124 (59.6)	13.4 (5.58)		65.2 (11.49)	
Employment	Yes	102 (49.0)	14.4 (5.21)	1.219	62.6 (10.68)	−1.910 *
	No	106 (51.0)	13.5 (5.34)		65.6 (11.95)	
Meeting PLHIV ^1^	Yes	4 (1.9)	13.8 (3.59)	−0.072	74.5 (9.71)	1.848 *
	No	204 (98.1)	13.9 (5.32)		63.9 (11.38)	
HIV education	Yes	96 (46.2)	15.1 (4.38)	3.156 **	63.3 (12.08)	−0.921
	No	112 (53.8)	12.9 (5.77)		64.8 (10.83)	
HIV testing	Yes	6 (2.9)	16.7 (6.38)	1.286	67.5 (3.73)	0.737
voluntary	No	202 (97.1)	13.9 (5.24)		64.0 (11.56)	
HIV testing non-voluntary	Yes	44 (21.2)	16.1 (3.58)	3.994 ***	65.9 (14.53)	1.148
	No	164 (78.8)	13.4 (5.51)		63.6 (10.43)	
Information	Yes	46 (22.1)	16.1 (4.05)	3.756 ***	62.6 (13.72)	−0.893
about HIV	No	162 (77.9)	13.3 (5.44)		64.5 (10.69)	
Sexual	Heterosexual	183 (88.0)	14.3 (4.89)	1.908 *	63.6 (11.34)	−1.788 *
orientation	Other ^2^	25 (12.0)	11.4 (7.22)		67.9 (11.50)	
Sexual	Yes	141 (67.8)	14.4 (5.14)	1.860 *	64.1 (11.82)	0.070
experience	No	67 (32.2)	13.0(5.47)		64.0(10.62)	

^1^ PLHIV: people living with HIV, ^2^ Other: homosexual, bisexual, asexual, etc.; * *p* < 0.10, ** *p* < 0.01, *** *p* < 0.001.

**Table 2 ijerph-17-08076-t002:** Percentage of correct answers per question related to HIV knowledge.

No	Question	Correct	
		*n*	%
1	Presence of healthy-looking person with HIV	167	80.3
2	Risk reduction by using a condom	161	77.4
3	HIV transmission through blood	161	77.4
4	HIV transmission through penetrative vaginal sex	155	74.5
5	HIV transmission through penetrative anal sex	154	74.0
6	Existence of female condom	147	70.7
7	Origin of new sexually transmitted HIV infections from people unaware of their HIV-positive status	141	67.8
8	HIV transmission through sperm	132	63.5
9	HIV transmission through vaginal fluids	122	58.7
10	HIV transmission by sharing food with people with HIV	115	55.3
11	HIV transmission through giving oral sex	115	55.3
12	Impact of sexually transmitted disease on the risk of getting HIV	111	53.4
13	HIV transmission through sweat	107	51.4
14	HIV transmission through mutual masturbation	102	49.0
15	HIV transmission through saliva	97	46.6
16	Risk reduction of HIV transmission by having sex with only one uninfected partner who has no other partners	97	46.6
17	HIV transmission through urine	93	44.7
18	HIV transmission through kissing	90	43.3
19	Post-exposure prophylaxis for HIV infection	89	42.8
20	Progression of HIV infection to AIDS	87	41.8
21	HIV transmission by mosquito bites	75	36.1
22	Gap in life expectancy between people with and without HIV	74	35.6
23	Duration that HIV infection can be detected after exposure	68	32.7
24	Prevention of mother-to-child transmission of HIV	67	32.2
25	Risk reduction in sexual HIV transmission for PLHIV ^1^ on antiretroviral treatment and responding to treatment	66	31.7
26	The most important route of HIV transmission in our country	55	26.4
27	HIV transmission through human bites and accidental needle stick injuries	51	24.5

^1^ PLHIV: people living with HIV.

**Table 3 ijerph-17-08076-t003:** Correlations among knowledge, subitems of stigmatizing attitude toward HIV, and subjective norms for SSB (*n* = 208).

Variable	1	2	2-1	2-2	2-3	2-4	2-5	2-6	2-7
1. Knowledge	-								
2. Stigmatizing attitude (total)	−0.111	-							
2-1. Concerns about occasional encounters	−0.219 **	0.613 **	-						
2-2. Avoidance of personal contact	0.084	0.701 **	0.400 **	-					
2-3. Responsibility and blame	0.039	0.732 **	0.189 **	0.519 **	-				
2-4. Liberalism	0.222 **	0.588 **	0.101	0.483 **	0.533 **	-			
2-5. Non-discrimination	−0.241 **	0.539 **	0.529 **	0.212 **	0.124	−0.050	-		
2-6. Confidentiality of serological status	−0.187 **	0.597 **	0.318 **	0.254 **	0.304 **	0.170 *	0.240 **	-	
2-7. Criminalization of transmission	−0.239 **	0.623 **	0.251 **	0.232 **	0.378 **	0.331 **	0.160 *	0.573 **	-
3. Subjective norms for SSB ^1^	0.188 **	−0.183 **	−0.107	−0.077	−0.107	−0.010	−0.127	−0.199 **	−0.208 **

^1^ SSB: safe sexual behavior.; * *p* < 0.05, ** *p* < 0.01.

**Table 4 ijerph-17-08076-t004:** Linear and quantile regression results for knowledge and stigmatizing attitude about HIV infection (*n* = 208).

Dependent Variable	Q ^1^	Independent Variable	Category	Coef. ^2^
Knowledge	OLS ^3^	Non-voluntary HIV testing	No (ref.)	
			Yes	1.87 *
		Exposure to HIV-related information	No (ref.)	
			Yes	2.37 **
		Experience of HIV education	No (ref.)	
			Yes	1.17
		Subjective norms for SSB ^4^	-	0.30 *
	q25	Non-voluntary HIV testing	No (ref.)	
			Yes	2.02
		Exposure to HIV-related information	No (ref.)	
			Yes	2.66 *
		Experience of HIV education	No (ref.)	
			Yes	1.09
		Subjective norms for SSB ^4^	-	0.23
	q50	Non-voluntary HIV testing	No (ref.)	
			Yes	1.33
		Exposure to HIV-related information	No (ref.)	
			Yes	1.51*
		Experience of HIV Education	No (ref.)	
			Yes	2.56 **
		Subjective norms for SSB ^4^	-	0.18
	q75	Non-voluntary HIV testing	No (ref.)	
			Yes	1.75
		Exposure to HIV-related information	No (ref.)	
			Yes	2.29 *
		Experience of HIV Education	No (ref.)	
			Yes	0.07
		Subjective norms for SSB ^4^	-	0.21
Stigmatizing attitude	OLS	Meeting PLHIV	No (ref.)	
			Yes	10.38
		Subjective norms for SSB ^4^	-	−0.61
	q25	Meeting PLHIV	No (ref.)	
			Yes	18.15 *
		Subjective norms for SSB ^4^	-	−1.17 **
	q50	Meeting PLHIV	No (ref.)	
			Yes	10.69
		Subjective norms for SSB ^4^	-	−1.00 **
	q75	Meeting PLHIV	No (ref.)	
			Yes	6.76
		Subjective norms for SSB ^4^	-	−0.80

^1^ Q: quantile, ^2^ Coef.: coefficient, ^3^ OLS: ordinary least squares, ^4^ SSB: safe sexual behavior; Note: In knowledge analysis, adjusted for age, marital status, education level, economic status, job, sexual orientation, and sexual behavior experience. In stigmatizing attitude analysis, adjusted for age, marital status, education level, economic status, job, sexual orientation, sexual behavior experience, and knowledge about HIV; * *p* < 0.05, ** *p* < 0.01.

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
