# Peer review of "Factors Influencing Young Korean Men’s Knowledge and Stigmatizing Attitudes about HIV Infection"

_ijerph, 2020, doi:10.3390/ijerph17218076_

Round 1

Reviewer 1 Report

This paper addresses important issues regarding the knowledge and stigmatising attitudes held by young Korean men. Overall, the paper contributes some important knowledge to the field, however, could be improved in some key areas.

The final paragraph of the introduction refers to specific sociodemographic characteristics that have been identified as influencing HIV-related knowledge and attitudes, however, these are not listed or referenced here. Some international research in this space would help strengthen this section (see, for example, Broady, T.R., Brener, L., Cama, E., Hopwood, M., & Treloar, C. (2020). Stigmatising attitudes towards people who inject drugs, and people living with blood borne viruses or sexually transmissible infections in a representative sample of the Australian population. PLoS One, 15(4), e0232218. doi:10.1371/journal.pone.0232218).

Study design and participants – I assume inclusion criteria also includes living in Korea (or something similar)?

HIV-related stigmatising attitudes – this section refers to the original tool and the one used in this study. Was it adapted for this study? Or does this refer to the Cronbach’s alpha score from each study (with both using the same measure)?

Please clarify the difference between receiving HIV-related education and exposure to HIV-related information. Could these be confounding variables in the analyses?

Data collection – was the token of appreciation a financial payment?

Results – presenting the range, mean, and SD of scale scores in isolation does not really contribute anything to the paper. Without a reference or comparison point, Table 1 does not provide any useful information (whereas the comparative scores in Table 2 are much more informative).

Table 3 would be easier to make sense of if questions were arranged according to the percentage who responded correctly.

I found the description of regression results difficult to follow and needed to read it a few times to make sure I understood what was presented. These findings are important (and the main contribution of the paper), so I suggest some additional editorial work to present these as clearly as possible.

How were the independent variables reported in Table 5 selected? In particular, I would be interested in the relationship between HIV knowledge/education and stigmatising attitudes.

The discussion talks about lack of HIV-knowledge as a major factor for negative HIV attitudes (paragraph 2), but then does not link the literature to current findings.

Stating that exposure to information was associated with increased knowledge is not surprising. More nuanced discussion around the relationship between these (and other ) variables is warranted.

Minor points:

The sentence structure, grammar, and use of tense should be revised throughout.

Ensure all acronyms are explained at their first use.

Author Response

Oct 30, 2020

Title: Factors Influencing Young Korean Men’s Knowledge and Attitudes About HIV Infection (IJERPH-958736)

Dear Reviewer,

Thank you for taking the time to review our manuscript. We have revised the manuscript according to your feedback and hope it is satisfactory. Please see the attachment.

Should you have any questions or comments, please do not hesitate to let us know.

Sincerely,

The authors

Reviewer 2 Report

Factors Influencing Young Korean Men’s Knowledge 2 and Stigmatizing Attitudes About HIV Infection

Decision: Accept with major changes

 Introduction is relevant, and the HIV stigma was clearly defined. The introduction has many weaknesses. The introduction could be strengthened by presenting the information from the discussion into the introduction. For example, how does the Korean society receive education about HIV transmission and prevention? Furthermore, the introduction did not mention anything about the prevalence of the homosexual community in Korea. How does the Korean society perceive the LGBT community? The authors state that 44% of new HIV infections are transmitted through homosexual contact. However, the introduction does expand on this population. 

Line 51 states KCDC. Could you explain the acronym, please?

Methods: There are significant concerns and questions regarding the conceptualization of this manuscript.

The authors failed to mention that the questionnaires were translated into Korean. The authors acknowledged this limitation; however, it is recommended to state it into the methods. 

The information presented in the data collection is limited and vague (Lines 125-128). For example, the authors state that “Data were collected during May 17–June 8, 2018, via online self-report questionnaire survey.” This section could be strengthened by mentioning how the participants were recruited. For example, were these participants recruited from a clinic or what type of website was it? How reliable is the data collected from an online website?

Additionally, the authors failed to mention if minors were included or excluded from the study. How did the authors assure minors were excluded from this study? The way this study is presented, anyone could take the online survey, and the data's accuracy could not be verified. This represents a major concern for this study.

Results: These are some major concerns in this section.

The authors did not present if any of their participants were homosexual. It is essential to state the number of self-declared participants who identified themselves as gay since the authors mentioned it in the discussion. However, there was no additional information.

The authors failed to present their instrument as an appendix.  

Discussion is relevant; however, there are some weaknesses.

For example, the discussion should start with a concise summary of the results and ensure the reader understands the total number of participants in this study (n=208). Additionally, the authors mention the CDC guidelines for the US lines 252-256 "In 2006, the U.S. Centers for Disease Control and Prevention (CDC) recommended integrating HIV testing into routine medical "

The authors failed to mention how the CDC guidelines are used in Korea. Shall the reader understand that the CDC issued guidelines are used for prevention in Korea, and the public health arena uses these guidelines when decisions are made? Furthermore, these guidelines are out of date. Another major concern in the discussion is the lack of mentioning of "U=U" statement. Does Korea use it as well?

The authors failed to recommend future studies based on the limitations of their research.

Conclusions are good. However, the authors state some recommendations. It is recommended to move this section at the end of the discussion or keep it in the conclusion and state Conclusions and Recommendations. (Lines 327-338) “Therefore, it is necessary to develop intervention programs providing more access to HIV-related information and education to improve the level of HIV-related knowledge among Korean  men in their 20s and to provide policy support at the institutional level to encourage the general population to undertake HIV screening. When offering interventions to reduce stigmatizing attitudes, particular attention should be given to choice of contents and methods to prevent indirect learning of stigmatizing attitudes; this should include providing opportunities for positive personal meetings with PLHIV and ensuring accurate perception of the HIV-positive status. Thus, the results of this study can provide basic data for future work aiming at interventions to improve HIV-related knowledge and reduce the stigmatizing attitudes among Korean men in their 20s, who show widely varying levels of HIV-related knowledge and HIV stigma.”

Author Response

(The authors gave the same response as above.)

Round 2

Reviewer 2 Report

The authors addressed all my comments.